# Self-Organisation of Prediction Models [note 1]

**DOI:** 10.3390/e25121596

**Published:** 2023-11-28

**Authors:** Rainer Feistel

**Affiliations:** Leibniz Institute for Baltic Sea Research (IOW), 18119 Rostock, Germany; rainer.feistel@io-warnemuende.de

**Keywords:** symbols, models, information, prediction, decisions, causality, experience, ritualisation, kinetic phase transition, activity

## Abstract

Living organisms are active open systems far from thermodynamic equilibrium. The ability to behave actively corresponds to dynamical metastability: minor but supercritical internal or external effects may trigger major substantial actions such as gross mechanical motion, dissipating internally accumulated energy reserves. Gaining a selective advantage from the beneficial use of activity requires a consistent combination of sensual perception, memorised experience, statistical or causal prediction models, and the resulting favourable decisions on actions. This information processing chain originated from mere physical interaction processes prior to life, here denoted as structural information exchange. From there, the self-organised transition to symbolic information processing marks the beginning of life, evolving through the novel purposivity of trial-and-error feedback and the accumulation of symbolic information. The emergence of symbols and prediction models can be described as a ritualisation transition, a symmetry-breaking kinetic phase transition of the second kind previously known from behavioural biology. The related new symmetry is the neutrally stable arbitrariness, conventionality, or code invariance of symbols with respect to their meaning. The meaning of such symbols is given by the structural effect they ultimately unleash, directly or indirectly, by deciding on which actions to take. The early genetic code represents the first symbols. The genetically inherited symbolic information is the first prediction model for activities sufficient for survival under the condition of environmental continuity, sometimes understood as the “final causality” property of the model.

## 1. Introduction


*Für einen Organismus muß die Welt voraussagbar sein, sonst kann er in ihr nicht leben. (English: “For an organism the world must be predictable, otherwise it cannot live therein.”)*

*Irinäus Eibl-Eibesfeldt, 1998 [1]*



*The theory of life is a theory for the generation of information.*

*Manfred Eigen, 2013 [2]*


Life on Earth emerged by self-organisation. Following Eibl-Eibesfeld (1998) [1], the ability to predict is a necessary condition for life; no organisms are known without this ability. Forms of “honorary life” (Dawkins 1996) [3] such as human apparatuses that are part of human culture also belong to the realm of life (Donald 2008) [4]. If we include those, there exists no prediction outside that realm, such that prediction is also a sufficient condition for life. From this perspective, the self-organisation of prediction models is a process equivalent to the self-organisation of life. In contrast to the various chemical and environmental ingredients to the beginning of life, however, prediction may be understood as a merely physical technique, based on causality and natural laws, independent of any specific biological or biochemical details. Even more generally, the evolution of prediction models may be considered as a common physical principle behind rather distinct biological, social, technical, and scientific evolution processes.

According to Eigen (1971, 1976, 1982, 1994, 2013, Eigen and Schuster 1977) [1,5,6,7,8,9], life is a process of generation and accumulation of information by means of repeated trial and error. Figure 1 shows schematically a simple conceptual model of a trial-and-error system interacting with its outside world, representing this way also any arbitrary organism from its perspective of prediction. “The ability to learn and form memories allows animals to adapt their behavior based on previous experiences” (Botton-Amiot et al., 2023) [10].

Trial, in particular, random trial, is an elementary, precursory version of prediction. The self-organisation of prediction may be understood as the transition from blind trial to sophisticated forecast based on causal models. Conventional physical systems such as a heat engine do not possess any prediction abilities. Figure 2 shows a conceptual model of such an inanimate open physical system, possessing internal non-equilibrium dissipative structures and performing related processes, interacting across an interface with its environment.

A striking distinction between Figure 1 and Figure 2 is the one between *symbolic information* and *structural information* (Ebeling and Feistel 1994, Feistel and Ebeling 2011, Feistel 2017a, 2023) [11,12,13,14]; see Section 5 and Section 6. Entropy may serve as an example demonstrating the difference. By the “negentropy principle of information” (Brillouin 2013 [15]: p. 153), entropy is often described as a quantitative measure for the amount of information contained in a certain physical structure. Introduced empirically by Clausius (1865, 1876) [16,17] and statistically by Planck (1906, 1966) [18,19], thermal entropy is a measure of the amount of structural information (or physically bound information). Its value depends on the physical nature and on the state of a given object; for example, the entropy of a mass of liquid water is different from that of the same mass of ice, even at the same temperature and pressure (Feistel and Wagner 2006) [20]. The entropy proposed by Shannon and Weaver (1964) [21], by contrast, is a measure of the amount of symbolic information (or physically free information); it does not depend on the physical nature of the particular information carriers, be those neural nerve pulses, electronic computer bits, or ink-printed letters (Brillouin 2013, Feistel 2017a, 2019) [13,15,22].

“From the perspective of evolution theory, the world of sign-likes appears as a stage of evolution that was preceded by a world of not yet sign-likes“ (Nöth 2000 [23]: p. 135). “Semiosis is the process in which the sign (and meaning) emerges. In other terms, semiosis is interpretation” (Kull 2018 [24]: p. 455). The schematic transition process from Figure 2 to 1 is a self-organised replacement of a structural process by a symbolic process. At the transition point, which will be described here as a *ritualisation* transition, the two processes are actually identical, as is generally characteristic for the phase transitions of the second kind (Appendix A). Such a transition has occurred at the beginning of life, as will be considered in more detail in Appendix B. A similar transition has also happened in recent times in the technical world, such as the transition from cybernetic systems using mechanical or electrical feedback circuits and relays (Wiener 1948, Kämmerer 1974, 1977) [25,26,27], functioning as in Figure 2, to artificial intelligence, which is learning by the trial-and-error method of Figure 1, and is much more flexible through using symbolic information.

Clearly, in the end, any symbolic process is also some physical process, similar to the “naked” structural process, but, in the symbolic one, the physical aspect is not the essential contribution. If a task needs to be solved on a computer, this is performed physically by certain mechanical or electronic switches or relays, but the kind of (structural) hardware is not crucial for the task, while the (symbolic) software implemented on the hardware is the significant aspect for the solution of the problem. The arbitrariness of the particular hardware platform had been formulated by Turing (1950) [28] as the *universality principle* of digital computers. With respect to symbolic information processing, this principle implies the *code symmetry* (Feistel 1990, 2017a,b), [13,29,30] or the semiotic *arbitrariness* (Nöth 2000) [23], or the *principle of code plurality* (Kull 2007) [31] of symbols in representing a certain meaning. From the “symbolic inside world” of Figure 1, it is plausible that the functioning of the trial-and-error feedback loop does not depend on the specific kind of symbols that may be used within that world as its proprietary “internal affair”. This functioning may work as well with other structures operating as the symbols involved. Elsewhere, and also in a distant future, those actual symbols do not need to be understood.

In this paper it will be assumed that any symbolic information, in distinction to its structural counterpart, has a *purpose*, and that this purpose consists of its influence on the future decisions and physical actions taken by the receiver of the symbolic information. As a form of *final causality*, see Section 3, purpose means some intended or expected future result or aim of a present process or structure; as such, purpose is part of a prediction model. Purpose is something meaningless in the lifeless world (Ellis 2023) [32]. Symbolic information in its own right is futile; it obtains its relevance only after subsequent conversion to structural information within the associated information-processing context. In the understanding of this paper, models, and, in particular, prediction models, are special symbols themselves. The self-organisation of prediction models, suggested here as a transition between the models of Figure 1 and Figure 2, requires the emergence of symbols and requires the transition from the transfer of structural information to the transfer of symbolic information in repeated interaction with an outside world.

By humans or any other living beings, *decisions* made now will matter only later on in the future. Beneficial decisions require good prognoses. *Causal models* can exploit past experience to predict upcoming events or circumstances. By appropriate receptors, after suitable conversion to symbolic information, structural information received from the environment may be filtered and stored in suitable symbolic form. After having passed through a symbol-processing model, the symbolic result needs to be transformed back into structural information transmitted to the environment, such as by triggered mechanical activity, incarnating the actual decision.

Living organisms are self-organised dissipative structures. To stay alive and multiply, they need a permanent supply of high-valued energy to compensate the inevitable production and export of entropy, to assemble and accumulate energy-rich molecules that make up the body, as well as to supply internal energy stocks to be exploited for driving active behaviour. The latter is ruled by a series of decisions of what needs to be performed when and how, being permanently made by any living being, from the simplest single cell up to human life and labour. The future fate of an organism is affected by any decision derived from experience made in the past and triggered by suitably adjusted prediction models, estimating what is expected to come. Through trial and error, symbolically stored sensational experience is used to evaluate the success of previous decisions and to modify the prediction model accordingly.

The self-organisation of prediction models requires several qualitative steps, although not necessarily in this temporal sequence:(i)Symbols need to emerge from non-symbolic, structural information processing;(ii)Sensors need to emerge which convert received structural information into symbolic information;(iii)Experience in the form of symbolic information needs to be stored in memory;(iv)Symbols need to be combined in networks to form symbol-processing models;(v)Symbols produced by models represent the evaluation result of the processed experience;(vi)Decision-making models convert symbolic values back into structural information of activity.

In natural evolution, this process is typically rolled out from the end, starting merely by spontaneous, uncontrolled activities such as volume growth or excretion of metabolic residues. Internal information processing has developed and advanced any already existing activities to become more and more diverse, sophisticated, energy saving, effective, and beneficial with respect to survival. Simple organisms perform certain mechanical or chemical activities without recognising any environmental signals. “Lower animals often possess a richer embodiment of their activity system as compared to a poor perception system” (von Uexküll 1973 [33]: p. 161). For example, comb jellies are the oldest known animals with a nerve net (Schultz et al., 2023; Moon et al., 2023) [34,35]. Likely, such first nervous systems had developed for active body motion to be performed synchronously in space and time, coordinated by symbolic nerve pulses, yet, assumingly, without the reception of external signals and lacking any symbolic processing of such perception. However, even in the absence of a centralised nervous system, the closely related starlet sea anemone already possesses memory and prediction capabilities in the form of Pavlovian conditioning with respect to external structural information when being exposed to light as a stimulus (Botton-Amiot 2023) [10].

Subsequently, structural information from its surroundings, such as temperature or brightness, may affect the organism’s metabolism. If this enhances the fitness, direct physical impact may develop into the specialised reception of selected external signals. Direct physical links (structural information) between receptor and effector may turn into more versatile symbolic information transfer by the ritualisation transition. Chemical symbols, such as specific indicator molecules, may be processed by logical gates such as NOT or AND, as this is known from properties of allosteric enzymes (Oubrahim and Boon Chock 2016) [36], similar to information processing in electronic computers. Networks of this kind may recognise signals of a certain duration rather than just instantaneously occurring conditions, that is, they may build up memory devices. Memory is formed from observation when “a rate-dependent [external] dynamical state is coded into quiescent [internal] symbols” (Pattee 2001 [37]: p. 5).

The paper is organised as follows. Prediction models emerge and work between sensual perception and the decided action of individuals, between the input of structural information converted to symbolic information, and the output of structural information after conversion from symbolic information. In Section 2, the terms “symbol” and “model” are specified and compared with other common similar words. In Section 3, the relation of causality and final causality to prediction models is discussed. The role of decisions, physical as well as symbolic decisions, as a transformer of symbolic to structural information is considered with simple physical examples, such as homoclinic orbits, in Section 4. Symbolic information is compared to structural information in greater detail in Section 5, and the self-organised emergence of novel symbolic information out of existing structural information by the ritualisation transition, as the key process for the self-organisation of prediction models, is characterised by selected contrasting properties in Section 6. The paper is discussed from a more general perspective in Section 7. To assist the reading, Appendix A reviews selected general properties of self-organisation processes and phase transitions. With respect to the origin of life, Appendix B explains briefly a conceptional ritualisation scenario. Appendix C reports selected mathematical properties of binary relations, graphs, and their adjacency matrices. Order and semi-order relations are briefly introduced in Appendix D, as well as groups and semi-groups in Appendix E.

## 2. Symbols and Models

Computer bits, feather colours, or printed letters are symbols. Words like “energy”, “entropy”, “information”, or “symbol” are also symbols. In the literature, in particular, in semiotics, symbols may also be regarded as “signs”, “icons”, “displays”, or “signals” (Oehler 1995, Deacon 1997, Nöth 2000, Pattee 2001, Feistel 2023) [14,23,37,38,39]. Within some external context, *symbols* are physical structures that represent something other than themselves, namely, the symbol’s *meaning*. The relation between the symbol’s structure and its meaning is arbitrary and assigned by convention (Nöth 2000, Lacková et al., 2017) [23,40]. Arbitrariness, that is, neutral stability with respect to fluctuations among any arbitrary suitable carriers, is the specific functional symmetry of symbol-processing systems (Feistel 1990, 2017a,b, Feistel and Ebeling 2016) [13,29,30,41]. Accordingly, the self-organised emergence of arbitrariness has properties of a kinetic phase transition of the second kind; see Appendix A. The corresponding fundamental character of this arbitrariness, of the purely conventional character of the relation between the physical structure of a symbol and its meaning, was proposed to be termed the “central dogma of semiotics” by Deacon (2021) [42], emphasising its key relevance.

*Models* do exist for the climate, for a steam engine, or for sailing vessels. Construction plans, cooking recipes, or genetic strands are also models for the physical structures that appear through the execution of those symbolic instruction sets. Some authors understand models as opposed to theories; this is not so here. Following Stachowiak (1973 [43]: p. 56), “a model is likewise … the most elementary item of perception as well as the most complex, most comprehensive theory”. “The word ‘model’ … is used … to mean an approximate description of an aspect of reality, with this description being developed for a specific purpose“ (Willink 2013 [44]: p. 16).

Models represent something other than they physically constitute in their own right. This property specifies models to be a special class of symbols. Typically, models are complex, consisting of structured sets of simpler, more elementary symbols. Similar to symbols, which may also represent other symbols rather than directly any physical reality, models may also represent other symbols or models. The text of this paragraph, for example, is a model of a model, similar to any other scientific article which consists of ordered sets of symbols (letters, words, numbers, figures) representing the research object, be that an observed or measured physical structure or another model (theory, hypothesis, simulation). Similarly, the notion of “entropy” is a model for certain fundamental properties of a macroscopic physical object, rather than being any kind of real physical “substance” itself. Sets of mutually consistent models are not necessarily pairwise reducible to one another, as if they were forming, in this way, a connected group or semigroup of models. Irreducible such models are often described as “emergent models”, “emergent quantities”, or “emergent properties” (Butterfield 2012, Fuentes 2014, Feistel and Ebeling 2016) [41,45,46].

A particularly important group of models is that of *mental models* (Craik 1943) [47] implemented in the brains of higher animals, especially of humans. Mental models result from the combination of phylogenetic (inherited) experience and ontogenetic (individually undergone) experience. “The mental model is the arena where imagination takes place. It enables us to experiment with different scenarios by making local alterations to the model” (Pearl and Mackenzie 2019 [48]: p. 26). Highly relevant for physicists and philosophers is the human model of *naïve realism* (Born 1965a, b) [49,50]. “The reality of a simple, untaught human is what he/she [immediately] feels and recognises. … The reality of those things which surround him/her is self-evident to him/her. … This attitude is termed naïve realism. The large majority of humans remains with that” (Born 1965a [49]: p. 53, 54). “Naïve realism is a natural attitude expressing the biological situation of humans and all animals” (Born 1965b [50]: p. 106). Naïve realism is a self-organised mental model as the result of successful Darwinian survival of all ancestors in the past (Hoffman 2020, Feistel 2021, 2022, 2023) [14,51,52,53], rather than an *a priori* principle of human understanding (Kant 1956) [54], possibly of divine origin.

Most models serve as prediction models, directly or indirectly. In symbolic form, they provide estimates for expected future observations, derived from similar experience already stored symbolically in memory, in combination with recent input, such as sensation suitably converted to symbols (Figure 1). In turn, observationally successful predictions serve as criteria for the reliability of the responsible model, to be used again in the future upon the repetition of similar sensations. By repetition, models accumulate information about the properties of the represented object, such as the real outside world. Causality, as a hypothetically lawful link between repeatedly observed correlated events, is an established construction principle for empirical prediction models. “Only after an activity has been performed and therefore belongs to the past, are we entitled to an attempt of understanding it from the perspective of pure causality” (Original text: “Erst wenn eine Handlung vollzogen ist und somit der Vergangenheit angehört, sind wir zu dem Versuch berechtigt, sie von rein kausalem Gesichtspunkten aus zu verstehen“) (Planck 1937 [55]: p. 29).

Causal models are the most successful prediction tools. “The validity of the causal law is connected with the possibility of making correct predictions for the future” (Planck 1948b [56]: p. 3). “If it is the task of science to look for lawful relations in all what happens in Nature and in human life, an inevitable prerequisite for that is … that such a relation in fact exists, and may be described in clear words. In this sense we tend to talk about the validity of a general causal law and about the determination of all processes in the natural and the mental world by this law. However, what does it mean that a process, an event, an activity occurs with lawful necessity, is causally determined, and how can the lawful necessity of a process be detected? I have no better idea to provide a clearer and more convincing proof for the necessity of a process than by the possibility of predicting the occurrence of the particular process” (Original text: “Wenn es die Aufgabe der Wissenschaft ist, bei allem Geschehen in der Natur oder im menschlichen Leben nach gesetzlichen Zusammenhängen zu suchen, so ist … eine unerläßliche Voraussetzung dabei, daß ein solcher Zusammenhang wirklich besteht, und daß er sich in deutliche Worte fassen läßt. In diesem Sinne sprechen wir auch von der Gültigkeit des allgemeinen Kausalgesetzes und von der Determinierung sämtlicher Vorgänge in der natürlichen und in der geistigen Welt durch dieses Gesetz. Was heißt nun aber: ein Vorgang, ein Ereignis, eine Handlung erfolgt mit gesetzlicher Notwendigkeit, ist kausal determiniert, und wie stellt man die gesetzliche Notwendigkeit eines Vorganges fest? Ich wüßte nicht, wie man für die Notwendigkeit eines Vorganges einen deutlicheren und überzeugenderen Nachweis erbringen kann als dadurch, daß die Möglichkeit besteht, das Eintreten des betreffenden Vorgangs vorherzusehen.”) (Planck 1937 [55]: p. 5).

Sudden deadly risks cannot be learned by individual ontogenetic experience because the killed organism cannot store this information in its memory for later. Organisms, however, which, due to randomly modified prediction models, instinctively avoid related risky situations, can inherit their survival strategy as phylogenetic experience. Such warnings may appear emotionally as a diffuse “fear” or “phobia” without clear causal justification. It seems that, for this reason, many big predators are afraid of humans and tend to avoid contact (Brown et al. 1999, Smith et al. 2017, Gaynor et al. 2019, Zanette et al. 2023) [57,58,59,60]. This indirect feedback mechanism is related to the psychological phenomenon of biased recognition known as “silent evidence” (Taleb 2008) [61]. When, after an earthquake, a few survivors praise their god for saving their lives, while the many killed victims fail to oppose, then the earthquake finally appears as a convincing reason to trust in god. Winners write history. Prediction models may exploit information about never-experienced events.

Symbols, and especially, models, have two conjugate aspects: on the one hand, the way they emerge by self-organisation, and on the other hand, the way they are used in systems processing symbolic information (Feistel 2023) [14]. The first aspect may be denoted as the *design time* of a symbol; this process is described here as the *ritualisation* transition. The second aspect may be denoted as the *run time* of a symbol, which is denoted as a *symbolisation* process that takes place, e.g., during an observation or a measurement that extracts symbols such as nerve pulses or measured numbers from the structural information of the given external object or measurand. “Measurement is a form of symbolisation. It consists in assigning numerals to objects or quantities” (Craik 1943 [47]: p. 75).

## 3. Causality and Finality

Although causality cannot be perceived in nature (Hume 1758, Russel 1919) [62,63], it is an extremely useful concept for the construction of prediction models, especially of human mental models (Kant 1956, Planck 1948b, Feistel 2023) [14,54,56]. The physical concept of causality is a strictly irreversible one (Prigogine 2000, Riek 2020) [64,65]: a cause always precedes its effect in time. In the literature of philosophy, biology, and semiotics, however, a “final causality”, or “finality”, or “retrocausation” is also extensively discussed as the phenomenon by which the final result of a process is actually assumed to be its “cause” (Sapper 1928, Nöth 2000, Nomura et al. 2019, Pink 2021, Deichmann 2023) [23,66,67,68,69]. Actually, prediction models may provide a logical link between the two disjunct causalities.

Why are there symbols at all? Darwinian selection demands the use of prediction models by competitors in order to gain selective advantage. In turn, causal prediction models require the prior emergence of symbols. In this sense, the “purpose” or “final cause” for the existence of symbols and for the self-organisation in the form of the ritualisation phenomenon is the need for prediction that arose from the possibility of prediction by gradually modifying random trial activities. Symbol processing makes prediction faster, energetically cheaper, more effective, more reliable, and more flexible, similar to digital technology as compared to its analogue forerunner. Concerning systems that are equipped with an appropriate prediction model, finality is consistent with causality. However, the system’s future state is not controlling and “causing” the system’s development, but, rather, the inherited prediction model is attempting to repeat the previous success of its predecessor’s mature structure and processing. This successful repetition is possible under the requirement of environmental continuity and of persistent boundary conditions. If, otherwise, the system’s external conditions change so quickly and dramatically beyond some critical tolerance limit, the system will fail to achieve the expected mature state because the prediction model becomes unable to properly predict the result of the development under the altered boundary conditions.

Darwinian evolution relies on such a continuity principle (Feistel 2023) [14]: “The world must be predictable for an organism to live therein” (Eibl-Eibesfeldt 1998 [1]: p. 21). “For any form of life, from unicellular organisms to large-brained mammals, living in a predictable environment is essential for increasing its chances for survival. … Anticipatory acts or predictive behavior are prerequisites for living organisms to sustain their survival when escaping from a predator, catching prey, or schooling” (Nomura et al. 2019 [67]: p. 267). If the parental genetic survival recipe suddenly becomes inappropriate to also ensure offspring survival because of environmental discontinuity, evolution cannot take place through trial and error and through the gradual accumulation of symbolic information in successively improving prediction models. While, say, cyanobacteria have a wide tolerance range to survive under strongly varying conditions, highly specialised species such as dinosaurs or humans run a higher risk of extinction. The pace by which the global human population is currently overturning the terrestrial ecosystem has become intolerably fast for numerous other recent species; they can no longer adapt their genetic prediction model through the traditional mutation-and-selection mechanism. Final causality as fitness for purpose may not function in those cases.

Evolution has established *deliberation* as a prediction method of mammals, which permits quick reactions and decisions about their immediate activity under circumstances never experienced before (LeDoux 2021) [70]. Evolution has established *sex* (Smith 1988, Margulis 2017) [71,72] as the most successful method to survive unpredictable situations, especially during population bottlenecks, by keeping available a variety of alternative genetic prediction models. Sexual reproduction and selection gave rise to the evolution of a spectacular wealth of new symbols used in mating activities (Darwin 1859, Prum 2017) [73,74]. The price to pay for the benefits of sex, however, is the individual death (Margulis 2017) [72].

Physical causality is an asymmetric binary relation between certain pairs of events. “The temporal sequence, however, is the only empirical criterion of the effect in relation to the cause which precedes the effect” (Kant 1965 [54]: p. 253 of edition A). The time evolution of reversible processes may be described mathematically by a group while irreversible causal processes satisfy semi-group properties (Appendix E). For a network of events, causality represents a mathematical semi-order relation rather than an order relation as each pair of events is not necessarily mutually linked, see Appendix C and Appendix D. If events are represented by nodes and their causal links by arrows (Greenland and Pearl 2017) [75], a causal network may be described using a directed graph or a non-negative adjacency matrix (Frobenius 1912, Harary et al. 1965, Lancaster 1969, Gantmacher 1971, Ebeling and Feistel 1982, Bornholdt and Schuster 2003) [76,77,78,79,80,81]. Final causality, however, if understood as a cause appearing later than its effect, violates the semi-order model of physical causality. Similarly, in fictitious time travel of science-fiction films and novels, the heroes carry in their brains back to the past also their individual mental prediction models along with their stored ontogenetic experience, so that their memory can symbolically “remember the future”. This is inconsistent with the causality’s semi-order properties and implies logical contradictions.

Likely, prediction models are physical systems that causally combine and connect symbols of events by semi-groups in a similar way as those had been observed of real, structural events. It may be assumed that the expected sequence of symbolic events in the model is represented also by an irreversible structural process, percussing previous experience in a simplified form (Hertz 1894 [82]: p. 1).

## 4. Physical and Symbolic Decisions

A popular example for a decision is the millennium-old parable of “Buridan’s Ass”, named after the French philosopher Jean Buridan: a donkey placed exactly amid two equal stacks of hay is unable to decide on one and will eventually die of hunger. Mathematically, the donkey is located at a saddle point of a fictitious “surface of happiness”, with two equal maxima to the left and right. Tiny fluctuations may suffice to break this symmetry at the initial unstable steady state and to trigger a decision toward one of the heaps.

*Decisions* play a key role in human personal and social life. Back to Adam Smith (1776) [83], disciplines such as “decision theory” or “best-choice theory” typically investigate problems related to the reasons for, and consequences of, individual decisions in society. In the case of human decisions, those are often regarded as “free will” (Planck 1937) [55], and are widely and controversially disputed in the literature (Pauen and Roth 2008, Pink 2009, Maldonato 2014, Pearl 2022) [84,85,86,87].

However, decisions, in the particular sense as specified below, are more fundamental acts for biology in general than just those of humans. “Decision theory provides a means to find the optimum response given uncertain information by weighing appropriately the costs and benefits of each potential response” (Perkins and Swain 2009 [88]: p. 1). Prediction models are built to deliver symbolic information which leads to decisions through evaluating and comparing the expected benefits; asking for the physical roots of the self-organisation of such models implies the question of the physical roots of the self-organisation of decisions. Here, we shall discuss certain aspects of *physical decisions* and of *symbolic decisions*, assuming that those represent a physical basis of biological behaviour from the very beginnings up to human interests and related activities. The importance of decision (or choice) for symbolic information processing (or semiotics) in combination with prediction, experience, and memory has already been emphasised previously by Kull (2018) [24], who quoted Viktor von Weizsäcker’s (1940 [89]: p. 126) statement that “the process of life is a decision rather than a succession of cause and effect” (Original text: “Der Lebensvorgang ist nicht eine Sukzession von Ursache und Wirkung, sondern eine Entscheidung.”). “By ‘semiosis’ we mean the process of choice-making between simultaneously alternative options” (Kull 2018 [24]: p. 454).

“There is a large number of new phenomena which are associated to irreversibility, and appear only in systems far from equilibrium. … In front of a bifurcation, you have many possibilities, many branches. The system ‘chooses’ one branch” (Prigogine 2000 [64]: p. 5). When a straight elastic column is put under pressure, beyond a critical load it will suddenly bend, a phenomenon known as “Euler buckling” (Zeeman 1976 [90]). In the simplest case, the column has two options, to bend to the left or to the right. The decision to which side to bend depends on small fluctuations, asymmetric structures, or boundary conditions. Typically, the physical process initiating the decision occurs at an energy level much lower than that of the amplified processes that “explode” due to feedback processes in an accelerated manner after the decision was made. In this sense, a physical decision is a macroscopic amplification of a chosen option out of a microscopic manifold of those. This exponential growth is characteristic for the vicinity of unstable or metastable states (Haken 1977 [91]). For the special case of human decisions, “in order to understand ambiguity of perception and the occurrence of meaning, we treat the brain as a synergetic system, or, in other words, as a self-organizing system close to instability points” (Haken 1995 [92]: p. 23).

Let a *physical decision* be a macroscopic process triggered by a microscopic event, such as an avalanche released by a tiny tremor, a bomb exploding after a slight touch of its detonator, or the sudden freezing over of a supercooled liquid after a local thermal fluctuation. The fertilisation of an egg cell to form a zygote is a physical decision to start pregnancy. A physical decision is an apple that suddenly falls from a tree, or a spark igniting a wildfire. Physical decisions occur at unstable or metastable states (Summers 2023) [93]; they are irreversible and produce entropy.

An instructive dynamical model for a system capable of physical decisions is a “homoclinic orbit” (Shilnikov 1969, Gaspard et al. 1984, Drysdale 1994) [94,95,96]. “Shilnikov homoclinic orbits are trajectories that depart from a fixed saddle-focus point … and return to it after an infinity time” (Medrano et al., 2005 [97]: p. 1). If several different homoclinic orbits start from the same stationary point, a system may, by microscopic fluctuations, “decide” which orbit to follow, performing the orbit’s macroscopic dynamics as an “activity”, before returning asymptotically to the initial waiting state. The simplest model for such a decision is a saddle-type homoclinic orbit (Drysdale 1994) [96] with two leaves representing alternative decisions and performed activities.

A simple saddle-type homoclinic orbit is shown in Figure 3. A related possible system of canonical-dissipative equations (Graham 1973, 1981, Feistel and Ebeling 1989) [98,99,100] is
$$\frac{dx}{dt}=-\frac{\partial H}{\partial y},$$
(1)$$\frac{dy}{dt}=\frac{\partial H}{\partial x}-\gamma \left(H\right)\frac{\partial H}{\partial y},$$
with the functions $H\left(x,y\right)=y^{2}+{\left(x^{2}-a^{2}\right)}^{2}$ and $\gamma \left(H\right)=c{\left(H-a^{4}\right)}^{2}$. The dissipation inequality implied,
(2)$$\frac{dH}{dt}=-\gamma \left(H\right){\left(\frac{\partial H}{\partial y}\right)}^{2}\leq 0,$$
shows that the system possesses a homoclinic orbit (Figure 3) of the shape of a lemniscate, $H\left(x,y\right)=a^{4}$. The steady state at (0, 0) is a saddle point. Fluctuations in its stable directions let the system immediately return to the origin. Fluctuations in either of the unstable directions trigger macroscopic excursions, after which the system finally returns to the original state. Such a model may conceptionally reflect decisions such as those being made by an animal confronted with an enemy. The animal may either hide and do nothing, or decide to take the flight, or to attack.

Let a *symbolic decision* be a physical decision in the form of structural information that is triggered by symbolic information; see the following section. For example, for humans, the consequences of “speech acts”, which are “doing things with words” (Austin 1962, Bühler 1965) [101,102], are typical symbolic decisions. When, at her wedding, the bride declares “yes, I will”, then this symbolic message of just three words given to the audience will dramatically change her future social and personal life. When, in a market, some bargain ends with “deal”, then the offered goods will instantaneously exchange their owners and will face an altered fate. Such a deal is usually the result of a cost–benefit analysis performed by the mental prediction models of the participants. As an aside, the German word for “to exchange” is “tauschen”, a word that has common roots with “täuschen”, meaning “to deceive”, to mislead the opponent’s prediction model.

Mechanical switches used to start or stop engines are devices for making physical decisions, releasing significant amounts of energy upon a minor energetic effort such as pushing a button. Those become symbolic decisions as soon as the switch is operated electronically by symbolic computer bits rather than mechanically by human fingers. By virtue of its effect, a symbolic decision assigns a *structural meaning* to a symbol. While observation translates external structural information into internal symbolic information, Figure 1, the decision is the counterpart that translates symbolic information into the structural information of an action, physically affecting the external world. Symbolic information in its own right is useless; it gains its relevance only in connection with associated structural information that ultimately appears as a result of a symbolic decision. Such a “magic power” of symbols is marvelled in numerous legends and fairy tales.

## 5. Structural and Symbolic Information

Prediction models are used to transform available information about the past and present into yet unavailable information about the future. However, the meaning of the term “information” varies widely in the scientific and other literature. Here, information is a term used for special physical processes or structures, or for certain properties of those. *Information carriers* are structures requisite for any transfer, storage, and processing of information. There is no information without a physical carrier, quite in contrast to the understanding of some authors, who assume information to be the substance of which the world ultimately consists. With respect to the physical carrier structures, information itself is an emergent quantity, something assigned to those structures by an external context or agent. Occasionally, information itself is regarded as some physical quantity similar to entropy, as it may obey conservation and dissipation laws, and it may possess an *amount* and a *value*.

When receiving the honorary citizenship of Pescara in Italy, Ilya Prigogine (2000) [64] said in his inauguration speech, with respect to certain physical information theories, that “the pleasure of being invited to this beautiful ceremony, and my friendship with Professor Ruffini, would have been included in the information at the big bang. But that seems very strange, and I could never accept this view”. In this quotation, by “information”, physical *structural information* is meant that may (or rather, may not) had been preformed already at the big bang, in absence of any symbols, and has nonetheless been perfectly conserved from then on to the present day, and to any future yet to come. Such a putative conservation of structural quantum information, known as ”Hawking’s information paradox“, is subject to recent gravity research (Almheiri et al., 2020) [103]. It is well-known, however, that the macroscopic loss of information as a consequence of Clausius’ law of irreversibly increasing entropy is inconsistent with the microscopic reversibility of classical mechanics (Feistel and Ebeling 2016) [41], and neither can a similar inconsistency be excluded for quantum effects (Marletto et al., 2022) [104]. All these physical laws apply to structural information.

In this paper, structural information is rigorously distinguished from *symbolic information*. It is understood that any physical structure carries structural information just by its very existence. Symbols, the way they are introduced here, are physical structures which additionally carry symbolic information. The latter is assigned to the structure in the context of an external information processing system by convention, not reducible to the symbol’s intrinsic structural information. For example, a printed word carries a certain meaning as its conventional symbolic information that is independent of the word’s structural information, such as the kind of ink or paper used for printing. Symbols represent something other than themselves.

In solid-state physics, excitable modes that do not cost any energy to excite are known as “Goldstone modes” or “Goldstone bosons”, such as certain vibrational modes of the crystal lattice or magnetic spin waves. Such modes are sensitive to fluctuations and may modify the given structure without changing its energy. Typically, the phase of an autonomous oscillator like a pendulum is such a neutrally stable mode that is subject to random drift. For this reason, for example, clocks are never perfectly stable and deviate from one another over time. If a clock is paused for an arbitrary while, it will afterwards never return spontaneously to its unperturbed previous gait. In the linear stability analysis of dynamical systems, neutral Goldstone modes are associated with Lyapunov coefficients possessing zero real parts.

Physically, the “conventionality” or “arbitrariness” of the symbol’s meaning corresponds to such a Goldstone mode, expressed by a vanishing Lyapunov coefficient of the system’s dynamical equations with respect to fluctuations that replace the given structure with a different one with the same meaning (also see Step 6 in Appendix B). For example, if a text is typed in “Sans-serif” letter font and is replaced by the same text in “Arial”, the meaning of the text remains unaffected, and there is no physical restoration or relaxation force that tends to spontaneously return the text to “Sans-serif”. Similarly, all spoken languages are subject to natural “weathering” (Brunnhofer 1871, Müller 1872, Janson 2002) [105,106,107]; they drift regionally to different dialects and eventually diverge into new local languages.

The discovery of the planet Neptune may serve as an example for the relation between structural and symbolic information. The observational data of the planet Uranus published by Bouvard in 1821 deviated significantly from the values predicted by Kepler’s mathematical model. Le Verrier could explain the discrepancies mathematically by the existence of a yet unknown planet, Neptune, which could indeed be observed near the suggested position by Galle in 1846. This famous history can be understood in a way that the perturbation of Uranus’ orbit represents structural information that was transferred from Neptune to Uranus by gravity interaction. The existence and certain properties of Neptune are physically present in Uranus’ trajectory. By measuring Uranus’ motion, this information was converted to symbolic information in the form of numerical tables. By exploiting this experience, a mathematical prediction model then provided the hypothetical position of Neptune in the sky. Through the decision of pointing a telescope to that spot, the symbolic model result was converted back to structural information. The light observed from that star, again, as structural information, could be observed and converted again into symbolic information in the form of scientific communication about the new discovery, as a validation of the prediction.

Great apes can learn to use some words. They have never done that themselves, and have always been taught by humans. By contrast, nobody had ever taught early humans to use words to speak or write. The human use of language is definitely self-organised. In the course of the natural evolution of life, including humans, symbols emerged through self-organisation in a ritualisation process. Similarly, the emergence of the genetic code and the symbolic information it represents can be assumed to have occurred by self-organisation. A simple conceptional model for the origin of life understood as a ritualisation transition is briefly presented in Appendix B.

## 6. Properties of the Ritualisation Transition

Prediction models are special symbols; the self-organisation of prediction models became possible only along with the self-organisation of symbols. This relation makes *ritualisation*, as the key process for the emergence of symbols, a crucial event also for the evolution of prediction models. Ritualisation had previously been defined (Feistel 2017b) [30] to be the following:-“The gradual change of a useful action into a symbol and then into a ritual; or in other words, the change by which the same act which first subserved a definite purpose directly comes later to subserve it only indirectly (symbolically) and then not at all” (Huxley 1914) [108];-A process by which behavioural or physical forms, or both, that had originally developed to serve certain different purposes for communication within a population (Lorenz 1970) [109];-The modification of an animal behavioural pattern to a pure symbolic activity (Eibl-Eibesfeldt 1970) [110];-The development of signal–activity from use–activity (Tembrock 1977) [111];-The self-organised emergence of systems capable of processing symbolic information (Feistel and Ebeling 2011) [12].

Frequently, in the course of evolution, three stages of a ritualisation transition are observed: before, during, and after the transition (Feistel 1990, Feistel and Ebeling 2011) [12,29]. Initially, the existing structure is only slightly variable in order to maintain the system’s essential functionality. Successively, involved structures may gradually reduce to some rudimentary, simplified version of themselves, to “icons” or “pictograms”, which represent the minimum complexity requisite for the actual task (Klix 1980) [112]. As a kind of caricature, this may be a modified, a skeleton representation of the original structure in a way that emphasises some relevant characteristics and simplifies or omits other, irrelevant ones. Redundant partial structures are no longer supported by restoring forces, and related fluctuations may increase substantially. At the transition point, the “icons” turn into mere symbols that may be modified arbitrarily, thus expressing the emerging code symmetry, and permit divergent, macroscopic fluctuations, see Appendix A. As a result, the kind and pool of symbols may quickly enlarge and adjust to new external requirements or functions. Later, in a maturation phase, the code becomes standardised to maintain the intrinsic consistency and compatibility of the newly established information processing system. Fluctuations are increasingly suppressed, the code becomes frozen-in, and preserves, in its remaining arbitrary structural details, a record of its own evolution history. Such insights into the early stages of life are available from tRNA molecules which define the chemical meaning of genetic codons and are exceptionally stable throughout the phylogenetic history (Eigen 1982) [7]. For instance, in the ritualisation of spoken and written language, of numbers, or of gestures, all the above stages of evolution appear in a similar, more or less pronounced form (Feistel 2017a,b, 2023) [13,14,30]. “In all aboriginal languages, vestiges of these sounds of nature are still to be heard; though, to be sure, they are not the principal fibres of human speech” (von Herder 1772) [113,114].

Symbolic information has some general properties (Feistel and Ebeling 2011) [12]:(i)Symbolic information systems possess a new symmetry, the *carrier invariance*. Information may, loss-free, be copied to other carriers or multiplied in the form of an unlimited number of physical instances. The information content is independent of the physical carrier system used.(ii)Symbolic information systems possess a new symmetry, the *coding invariance*. The functionality of the processing system is unaffected by the substitution of symbols with other symbols, as long as unambiguous bidirectional conversion remains possible. In particular, the stock of symbols can be extended with the addition of new symbols or the differentiation of existing symbols. At higher functional levels, code invariance applies similarly also to the substitution of groups of symbols, synonymous words, or of equivalent languages.(iii)Within the physical relaxation time of the carrier structure, discrete symbols represent quanta of information that do not degrade and can be refreshed unlimitedly.(iv)Redundant copies of symbolic information may be carried along for error correction in cases of the loss or damage of the original.(v)Imperfect functioning or external interference may destroy symbolic information but only biological processing systems can generate new or recover lost information.(vi)Symbolic information systems consist of complementary physical components that are capable of producing the structures of each of the symbols in an arbitrary sequence upon writing, of keeping the structures intact over the duration of transmission or storage, and of detecting each of those structures upon reading the message. If the stock of symbols is subject to evolutionary change, a consistent co-evolution of all components is required.(vii)Symbolic information is an emergent property; its governing laws are beyond the framework of physics, even though the supporting structures and processes do not violate physical laws.(viii)Symbolic information is extracted from structural information by observation or measurement processes.(ix)Symbolic information has a meaning or purpose beyond the scope of physics, which becomes revealed by conversion to structural information, such as by symbolic decisions.(x)In their structural information, the constituents of the symbolic information system preserve a frozen history (“fossils”) of their evolutionary pathway.(xi)Symbolic information processing is an irreversible, non-equilibrium processes that produces entropy and requires a free-energy supply.(xii)Symbolic information is encoded in the form of the structural information of its carrier system. The source, transmitter, and destination represent and transform physical structures.(xiii)Symbolic information exists only in the realm of life.

Structural information has a number of different general properties (Feistel and Ebeling 2011) [12]:(i)Structural information is inherent to its carrier substance or process. Information cannot loss-free be copied to any other carrier or identically multiplied in the form of additional physical instances. The physical carrier is an integral constituent of the information, meaning and structure cannot be separated from one another. The state of the physical context of the system is an integral part of the information.(ii)There is no invariance of structural information with respect to structure transformations. Different structures represent different structural information.(iii)Structural information emerges and exists on its own, without being produced or supported by any kind of separate information source. No coding rules are involved when the structure is formed by natural processes.(iv)Over the relaxation time of the carrier structure, structural information degrades systematically as a consequence of the Second Law, and disappears when the equilibrium state is approached.(v)Internal physical processes or external interference may destroy structural information; it cannot be regenerated or recovered. Periodic processes can rebuild similar structures but never exactly the same, in particular because the surrounding world will never be exactly the same again at any later point of time.(vi)Structural information is not represented in the form of codes. No particular coding rule or language is required or distinguished to decipher a structure.(vii)Structural information is a physical property; it is represented by the spatial and temporal configuration of matter, and its governing laws are the laws of physics.(viii)Structural information is of a physical nature and is independent of life.

## 7. Discussion

Key aspects of seemingly unrelated scientific topics such as Leibniz’s (1765) [115] “final cause”, Hume’s (1758) [62] “scepticism”, Darwin’s (1859) [73] “natural selection”, Peirce’s “semiotics” (Nöth 2000) [23], Huxley’s (1914) [108]“ritualisation”, Born’s (1965a,b) [49,50] “naïve realism”, Prigogine’s (1969) [116] “dissipative structures”, Gilbert’s (1986) [117] “RNA world”, Pattee’s (2001) [37] “physics of symbols”, or Hoffman’s (2020) [51] “relative reality” may jointly be considered from a common perspective of self-organised prediction models. Active decisions governed by prediction models may be regarded as a universal property of life, independent of specific biochemical details or terrestrial conditions, and also include non-Darwinian forms of “honorary life” such as market economy and scientific or technological artefacts such as computers or artificial intelligence. The first prediction models emerged through self-organisation during the course of the coevolution of sensual reception, symbol processing and memory, and decisions on activities. Typically, self-organisation is characterised by the spontaneous formation of novel, “dissipative” structures or functions induced by symmetry-breaking kinetic phase transitions far from thermodynamic equilibrium, such as the oscillations of a hydrothermal geyser. Symbols may emerge in a similar manner through a universal kinetic transition that has been termed “ritualisation” in ethology, introducing arbitrariness as a new additional symmetry into information processing. In Appendix B, a conceptual model for the origin of life paints a simplified picture of the primordial ritualisation transition to the very first symbols and models. A similarly fundamental success of the self-organised emergence of symbols such as language and numbers, and of causal mental prediction models, is characteristic also for the historical ascent of humans. The variety of possible prediction models, from first genetic strands up to religions, technologies, and scientific theories, is subject to evaluation and selection through biological or social competition.

Illustrating certain aspects of this paper, here are some widely known but rather different examples related to selected features of prediction models:-**Phylogenetic experience***:* When Darwin (1859) [73] wrote his famous book “On the Origin of Species”, he mentioned, in his Chapter 1, various examples of the variability of phenotypic properties between parents and offspring: “When among individuals … any very rare deviation … appears in the parent … and it reappears in the child, the mere doctrine of chances almost compels us to attribute its reappearance to inheritance. … Perhaps the correct way of viewing … would be, to look at the inheritance … as a rule, and non-inheritance as the anomaly. The laws governing inheritance are for the most part unknown”. Despite that ignorance, only a few years later, Mendel’s (1866) [118] empirical inheritance rules went largely unnoticed by the scientific community. It took another century until Watson and Crick (1953) [119] as well as Nirenberg and Matthaei (1961) [120] revealed the molecular symbolic memory behind biological inheritance, known today as the “genetic code”. In this paper, genetic information is considered as an *inherited prediction model*, self-organised previously in the course of Darwinian selection by the long and unbroken track of successful ancestors, this way, keeping their accumulated *phylogenetic experience* available for their offspring as a predicted instruction set for the offspring’s subsequent survival and multiplication. This process may be regarded as the Darwinian evolution of prediction models in the sense of Dawkins’ (1976) [121] “selfish genes”.-**Ontogenetic experience**: When Pavlov, in 1905, measured the salivation of a dog in a lab, he noticed that, already, the sound of the walking technician started the dog’s mouth to water in expectation of the food the same person had always been providing. This classical conditioning (Denny-Brown 1928) [122] is controlled by a *mental prediction model* that had been established before by the repeated recognition of correlated events during the individual *ontogenetic experience* in the past. To make this happen, sensual impressions must be recorded symbolically in memory. Triggered by a repeated event, this information must be recalled and processed by the model in order to predict and await the yet missing events of the formerly observed scenario. “Brains are … essentially prediction machines” (Clark 2013 [123]: p. 181). The concept of mental models was developed by Craik (1943) [47].-**Scientific prediction laws**: When Clausius (1876) [17] studied the cyclic thermal processes of heat engines, he mutually compared numerous measured values of heat supply, dQ, at temperatures, T. He found that cycles with $\oint \frac{dQ}{T}<0$ are technically impossible: “Die algebraische Summe aller in einem Kreisprocesse vorkommenden Verwandlungen kann nur positiv oder als Grenzfall Null sein” (Clausius (1876) [17]: p. 223: English: “The algebraic sum of all transformations in a cyclic process can only be positive or, as a limiting case, zero.”). As a fundamental theorem, he concluded that “ein Wärmeübergang aus einem kälteren in einen wärmeren Körper kann nicht ohne Compensation stattfinden“ (Clausius (1876) [17]: p. 82, 364: English: “Heat transfer from a colder to a warmer body cannot occur without compensation.”). This “natural” law is a prediction model for, say, the maximum efficiency of any modern heat pump. Clausius (1865 [16]: p. 390, 1876: p. 94, 111) proposed a new thermodynamic state quantity, dS=dQ/T, termed “entropy” (“Verwandlung”, transformation, greek “τροπή”) by him (Feistel and Ebeling 2011, 2016) [12,41]. His most famous prediction was: “Die Energie der Welt ist constant. Die Entropie der Welt strebt einem Maximum zu” (Clausius (1865) [16]: p. 400: English: “The energy of the world is constant. The entropy of the world aspires to a maximum.”). Physical “natural” laws are symbolically formulated human models (Feistel 2023) [14], derived from past observations in order to predict the results of future observations or measurements.-**Observation–prediction–action cycle***:* Brahe’s meticulous observation of stars between 1586 and 1597 enabled Kepler to discover his pioneering laws of planetary motion, published in the books “Astronomia nova” of 1609 and “Harmonices mundi” of 1619. Kepler’s laws allowed successful predictions of the solar transits of Mercury in 1631 and of Venus in 1639, and later, even the discovery of Neptune in 1846. In 1687, Newton could demonstrate that his fundamental physical laws of bodily motion and of universal gravity were sufficient to correctly derive Kepler’s findings mathematically. In remote space regions never directly experienced by humans before, predictions by those laws gave rise to the first successful flight of an artificial celestial body, “Sputnik”, in 1957, confirming the merely symbolic predictions of astronomers in the form of structural information. Newton’s dynamical differential equations offer more comprehensive predictions than Kepler’s conservation laws of energy and angular momentum provide. “The ultimate goal of celestial mechanics [was] to resolve the great problem of determining if Newton’s law alone explains all astronomical phenomena” (Poincaré and Goroff 1993 [124]: p. I17).-**Causal prediction models***:* “Mathematically, the *law of causality* is expressed by the fact that physical quantities obey differential equations of a certain kind. The causal law of classical physics implies that the knowledge of the state of a closed system at some point of time determines its behaviour for all of its future” (Born 1966 [125]: p. 7). Causality is a key element of the human mental model of naïve realism. Causality does not exist in reality (Russell 1919 [63]: p. 180), nor can it be observed: “Through its sensational properties, no object may ever reveal the causes that produced it nor the effects that will result from it” (Hume 1758 [62]: p. 44). However, causality is an unrivalled human mental prediction tool (Orcutt 1952) [126]. The historical success of causal mental models made humans addicted to causal explanations for their personal observations, such as by superstition, religion, or science (Planck 1948a [127]: p. 23, Feistel 2023 [14]). “The human brain is the most advanced tool ever devised for managing causes and effects. … Causal explanations, not dry facts, make up the bulk of our knowledge” (Pearl and Mackenzie 2019 [48]: p. 2, 24). “We struggle for attributing cause and effect. Seeing events causally connected is an outstanding strategy to master our daily life“ (Mast 2020 [128]: p. 32).-**Mental prediction models**: The neuronally implemented, inherited prediction model of naïve realism emerged through self-organisation in the course of Darwinian evolution (Hoffman 2020, Feistel 2023) [14,51]. By introspection, Kant (1956) [54] painted a detailed picture of human naïve realism. Eighty years before Darwin (1859) [73], lacking a better explanation, Kant described causality as an a priori principle of reason rather than an empirical conclusion from phylogenetic experience. The alternative advantages either of exploiting intergenerational, phylogenetic experience stored in genetic information, or of fast and flexible individual, ontogenetic experience stored in brain memory, became combined through the socially distributed prediction models of science and technology of humans, permitted by the self-organisation of spoken and written language (Logan 1986, Pinker 1994, Deacon 1997) [39,129,130]. Sagan (1978 [131]: p. 39) regarded this kind of accumulated symbolic information as an “extrasomatic-cultural” one.-**Non-causal prediction models**: Scientific prediction models are not necessarily causal ones. For the description of technical or natural processes, for example, the quantitative knowledge of certain properties of physical objects may be required. Typically, a finite set of such properties is carefully measured and symbolically tabulated, similar to Brahe’s star-gazing, and subsequently represented mathematically by a continuous function, similar to Kepler’s and Newton’s laws, which predicts the properties under any other, not yet measured conditions. This way, as a special case, the properties of water, seawater, ice, and humid air are described in the form of empirical thermodynamic potentials by the international standard TEOS-10 [132], the “Thermodynamic Equation of Seawater—2010”, for use in numerical models for climate, oceanography, or desalination (IOC et al., 2010, Feistel 2018, Harvey et al., 2023) [133,134,135]. Such predicted property values should always be associated with estimated uncertainties (GUM 2008, Willink 2013, Feistel et al., 2016) [44,136,137]. The method of mathematical inter- and extrapolation, generalising locally observed situations to previously unexplored ones, is a powerful non-causal mental prediction tool that likely evolved from first geometric measurements in agriculture (Hilbert 1903) [138] and is still successfully applied in the latest science.

## Figures and Tables

**Figure 1 entropy-25-01596-f001:**
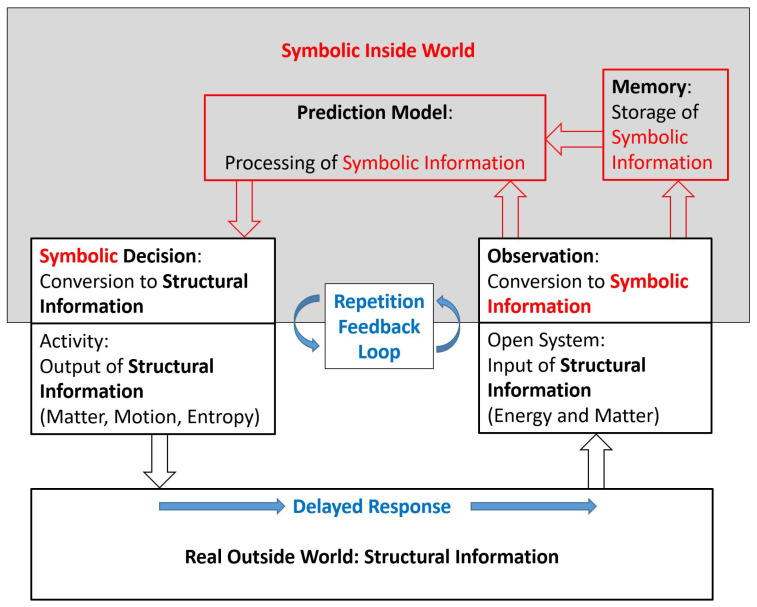
Schematic model of a self-organised trial-and-error system. Any activity performed by the system will structurally change its environment and, in turn, the way the latter becomes recognised. Sensors convert the actual perception into symbols such as nerve pulses or measurement values. Those symbols are stored and correlated with previous memory. A prediction model estimates future scenarios from the past experience. The preferred option among those predicted, still expressed in symbolic form, triggers an associated decision and is amplified to execute a subsequent activity, back in structural form. By repetition, successful predictability of the outside world permits the generation and accumulation of internal symbolic information about that world. The emergent symbolic information-processing subsystem is indicated by red colour. Structural information processing is indicated by blue colour.

**Figure 2 entropy-25-01596-f002:**
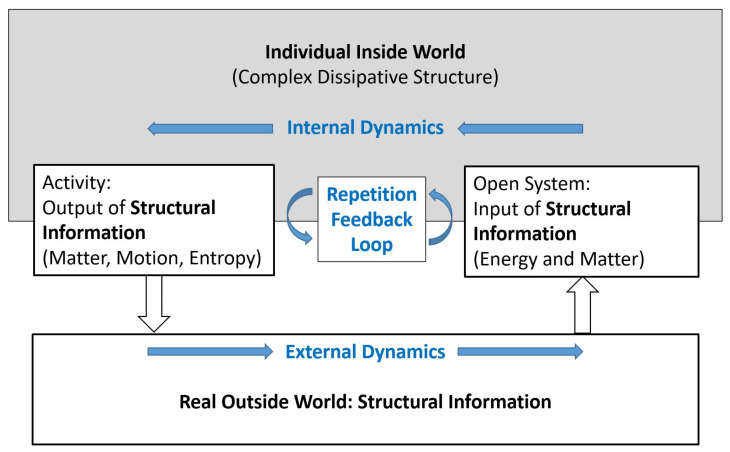
Schematic model of an open individual system interacting with its environment. As a dissipative structure, it necessarily releases its excess entropy to its surrounding. To maintain its structure, it needs supply of high-valued energy, typically in the form of energy-rich molecules or short-wave radiation. Interaction between individual and world may be understood as a continuous feedback loop of mutually receiving and transmitting structural information, modified by internal and external physical dynamics without symbolic manipulations. Structural information processing is indicated by blue colour.

**Figure 3 entropy-25-01596-f003:**
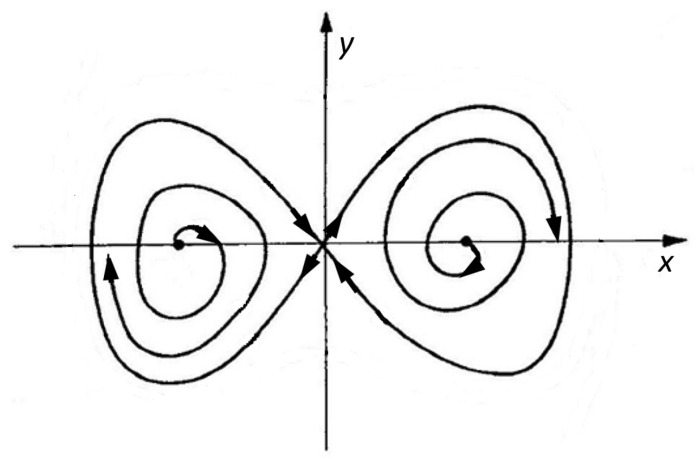
Schematic of a saddle-type homoclinic orbit as a model for a decision-making dynamical system.

## Data Availability

Not applicable.

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
