# Peer review of "Self-Organisation of Prediction Modelsâ€"

_entropy, 2023, doi:10.3390/e25121596_

Round 1
Reviewer 1 Report
Comments and Suggestions for Authors
This paper approaches the prediction process as a self-organising phenomenon.
First of all, the idea is novel and arouses great interest. Moreover, the introduction approaches the crucial problem of what living matter consists of in a convincing way.
On the other hand, identifying causal mechanisms as a fundamental ingredient of the problem is really interesting.
However, my perception when analysing this paper is that of reading a chapter of a popular book on complexity rather than a quantitative paper. For example, the use of anecdotes (lines 431-434, lines 457-469 etc.) is more typical of a long essay than of an Entropy paper. I wonder if the authors had this other format in mind when writing the article. It is in any case a thought-provoking article in every paragraph. Unfortunately, there are important shortcomings, which I quote below:
There is a totally excessive use of quotes and this makes for very heavy reading. The work almost seems like a succession of quotes which, although relevant and interesting, is excessive. I suggest that either many of them should be removed or that they should be much more synthesised.
Concerning the ideas of physical causality (lines 307-316) there is no mention of all the work done over decades by causality theorists such as Judea Pearl. On the other hand, it is a pity that the authors do not quantitatively develop the claim that causality represents a semi-group (indicating the expected associative properties, existence of neutral element and existence of opposite element).
Finally, I wonder why references to Hermann Haken's synergetics are omitted, since this author analyses in depth the phenomenon of perception and complexity.
Reviewer 2 Report
Comments and Suggestions for Authors
This paper provides a comprehensive review of the existing literature and offers valuable insights into the use of symbols, models, and the creation of prediction models. However, the theoretical framework for the self-organization of prediction models is somewhat rudimentary, and there may be some aspects where a broader perspective is needed. My specific observations are as follows:
1) While the paper touches on self-organization and discusses general characteristics of symbolic and structural information, as well as various instances of prediction models, it falls short of providing an in-depth exploration of potential approaches to developing a theory for the self-organization of prediction models. The paper should delve into the controversies and challenges associated with creating such a theory, highlight the current consensus mechanisms, and suggest potential research directions in this field.
2) The statement, 'Conventionality or arbitrariness of the symbol's meaning corresponds to a Goldstone mode,' would benefit from additional clarification and elaboration.
Round 2
Reviewer 1 Report
Comments and Suggestions for Authors
The authors have included all the suggested modifications and for my part there are no further recommendations.
Reviewer 2 Report
Comments and Suggestions for Authors
I have no further comments